# Suicidal Behavior and Club Drugs in Young Adults

**DOI:** 10.3390/brainsci11040490

**Published:** 2021-04-12

**Authors:** Giovanni Martinotti, Stefania Schiavone, Attilio Negri, Chiara Vannini, Luigia Trabace, Domenico De Berardis, Mauro Pettorruso, Stefano L. Sensi, Massimo Di Giannantonio

**Affiliations:** 1Department of Neuroscience, Imaging and Clinical Sciences, University G. d’Annunzio of Chieti-Pescara, 66100 Chieti-Pescara, Italy; giovanni.martinotti@gmail.com (G.M.); chiaravannini08@gmail.com (C.V.); mauro.pettorruso@unich.it (M.P.); ssensi@unich.it (S.L.S.); digiannantonio@unich.it (M.D.G.); 2Department of Clinical and Experimental Medicine, University of Foggia, 71122 Foggia, Italy; stefania.schiavone@unifg.it (S.S.); luigia.trabace@unifg.it (L.T.); 3S.C Area Ser.D Mantova—U.O. Ser.T Alto Mantovano, 46100 Mantova, Italy; ngrttl@gmail.com; 4Department of Clinical and Pharmaceutical Sciences, School of Life and Medical Science, University of Hertfordshire, Hatfield AL10 9AB, UK; 5NHS, Department of Mental Health, Psychiatric Service for Diagnosis and Treatment, Hospital “G. Mazzini”, 64100 Teramo, Italy; 6Center for Advanced Studies and Technology (CAST) University G. d’Annunzio of Chieti-Pescara, 66100 Chieti-Pescara, Italy

**Keywords:** suicide, suicide attempts, club drugs, polydrug abuse, novel psychoactive substances

## Abstract

Psychoactive drugs play a significant role in suicidality when used for intentional overdose or, more frequently, when the intoxication leads to disinhibition and alterations in judgment, thereby making suicide more likely. In this study, we investigated suicidality prevalence among drug users and evaluated the differences in suicide ideation, taking into account the substance categories and the association of suicide ideation intensity with other psychiatric symptoms. Subjects admitted to the Can Misses Hospital’s psychiatry ward in Ibiza were recruited during summer openings of local nightclubs for four consecutive years starting in 2015. The main inclusion criterium was an intake of psychoactive substances during the previous 24 h. The Columbia Suicide Severity Rating Scale (C-SSRS) was used to assess the suicide risk. Suicidality was present in 39% of the study cohort. Suicide Ideation Intensity overall and in the previous month was higher in users of opioids and in general of psychodepressors compared to psychostimulants or psychodysleptics. Suicidality was not correlated with alterations in any of the major psychopathological scales employed to assess the psychiatric background of the study subjects. The presence of high levels of suicidality did not specifically correlate with any major symptom indicative of previous or ongoing psychopathological alterations. These findings suggest that impulsivity and loss of self-control may be determinants of the increased suicidality irrespectively of any major ongoing psychiatric background.

## 1. Introduction

Epidemiological evidence indicates that the extent of substance abuse is not limited to patients with psychiatric disorders or substance use disorders (SUDs). On the contrary, the phenomenon mostly involves a heterogeneous cohort of users that includes “psychonauts,” clubgoers, students, marginalized or non-habitual recreational drugs consumers [1]. Moreover, recent studies show that the likelihood of approaching legal and illegal psychoactive substances occurs at an increasingly younger age. Substances are cheap and readily available online, thereby becoming accessible even to children, a phenomenon loaded with dramatic consequences in terms of mortality and psychiatric outcomes [2,3].

According to data from the U.S. study Adolescent Brain Cognitive Development (ABCD), after alcohol, cannabis is the second most used substance of abuse by adolescents with a prevalence rate of 35.6% [4]. A wide range of other substances is also available, including the so-called Novel Psychoactive Substances (NPS), like synthetic cannabinoids and synthetic cathinones, dissociatives, opioids, stimulants, gamma-hydroxybutyrate, and other “club drugs” [5].

In today’s globalized and fast-paced world, adolescents and young adults are constrained to an inner world dominated by “instantaneity.” Missing, among the youngsters, is the ability to ponder the medium and long-term effects of individual behaviors and life-choices as well the capability to project and envision themselves in stable social roles. The search for individualization, self-identity, intimacy, and future orientation is further complicated by societal pressure [6]. This psychosocial milieu leads users to seek instant gratification in a constant quest for euphoria, alertness, sociability, enhanced sensory perceptions, alterations of space and time perception, loss of inhibition, and improved sexual performance [7,8,9]. This behavior may be particularly at risk for adolescents and young adults who are often unaware of the health risks associated with the use of psychoactive compounds including those linked to the chemical contamination and adulteration by neurotoxic additives [5].

Acute and chronic drug misuse or abuse may impair judgment, weaken impulse control, and interrupt the functioning of critical inhibitory neurotransmitter pathways, thereby leading to enhanced suicidal tendencies driven by disinhibition [10]. Although few studies have addressed the issue of the incidence and predictors of suicide among users, psychoactive drugs play a significant role in suicidality when used for intentional overdose or, more frequently, when the intoxication leads to disinhibition and alterations in judgement that make suicide more likely [11,12]. For instance, a study reported that adolescent users of psychoactive substances are more likely to commit suicide by firearms than adolescents who do not use drugs [13].

Even the occasional use of psychoactive substances may be a significant risk factor for various dangerous conditions, including fatal and nonfatal overdoses, suicide attempts, and death by suicide [14,15]. Evidence indicates that, compared with the general population, individuals who use alcohol or drugs have a 10–14 times greater risk of death by suicide [16]. Furthermore, high rates of psychoactive substance use are found in post-mortem examination of body fluids of subjects whose death was classified as suicide or due to accidents [17]. This phenomenon is particularly alarming if one considers the novel, highly potent, and addictive compounds such as synthetic opioids flooding the market.

The dramatic rise in non-medical use of prescription opioids in the U.S. has been reported as an additional risk factor for intentional overdose and suicide [18,19]. In 2015, opioids were identified as the main cause of death in almost one-third of the suicides due to poisonings in the United States [20]. The epidemic is critically important in demographics that become less resilient to the risk of substance abuse due to increased chronic medical comorbidities, neurotoxic damage, and higher neuropsychiatric diseases [21]. Furthermore, as adolescence and young age involve a variety of major emotional, social, and physical changes, these factors are additional risk factors for the development of depressive symptoms and behavioral problems [17]. In this context, the risk of sudden death or suicide becomes exponential. In addition to this, several factors associated with substance use, such as psychiatric and medical comorbidities, financial difficulties, and unemployment, may also contribute to higher suicide risk [22].

The variety of medical and social consequences associated with substance use requires effective public health policies set to counteract these habits, as well as a continuous education update for health professionals to promote education and harm reduction [23,24].

Ibiza, one of the most popular nightlife hotspots for summer holidays in Europe, represents an important setting to search the psychopathological milieu of the misuse of novel and “vintage” drugs. Previous studies had evidenced a higher prevalence of risky behaviors showed by residents and tourists. These include problematic alcohol and substance use and the related intoxication, fatalities, overdoses, suicide attempts, or suicide [25,26].

Therefore, we conducted an observational study on a cohort of subjects who accessed the psychiatric ward of the Can Misses Hospital in Ibiza as a consequence of intoxication by psychoactive substances.

The main objective of the study was to assess psychopathological and suicidal features associated with substance use. Suicidality was measured considering three variables: suicidal ideation, suicide attempts, and death ideation according to The Columbia Suicide Severity Rating Scale (C-SSRS). The study also aimed explicitly to (1) identify the suicidality prevalence among drugs misusers, (2) evaluate the differences in Suicide Ideation Intensity (SSI; occurring through life and in the month before recruitment) among users of different substances, (3) evaluate the associations of SSI (lifetime and in the last month) with psychopathological features assessed by the Positive and Negative Symptoms Scale (PANSS), Mania Rating Scale (MRS), and Hamilton Depression Scale (HAM-D) scores.

## 2. Material and Methods

Subjects admitted to the psychiatry ward of the Can Misses Hospital in Ibiza were recruited for the study. The study was carried out during summer openings of local nightclubs (May–October) for four consecutive years starting in 2015. The inclusion criteria were (1) age 18–75 years old and (2) the intake of psychoactive substances or more than five alcohol units (e.g., 10 mL or 8 g of pure alcohol) during the previous 24 h.

Demographic and socioeconomic data, psychiatric and medical history, current pharmacological treatments, alcohol and substance use habits, including the use of NPS and prescription drugs. Psychiatric symptoms and conditions were evaluated at patients’ admission to the psychiatry ward according to the DSM-5 diagnostic classification. The first psychiatric assessment also included evaluating specific items related to suicidality, like (1) suicide attempts, (2) suicidal ideation, and (3) death ideation. A positive answer to at least 1 of these items was considered indicative of suicide risk for the patient.

An extended battery of psychodiagnostic tests was administered to patients during their hospitalization to explore the different psychopathological aspects of substance use. The battery investigated depressive or manic symptoms, anxiety, psychosis negative and positive symptoms, somatic disorders, aggressiveness, and suicidality. The battery included the Timeline follow-back for psychoactive substances and alcohol (TLFB); Brief Psychiatric Rating Scale (BPRS); Positive and Negative Symptoms Scale (PANSS); Mania Rating Scale (MRS); Hamilton Depression Scale (HAM-D); Hamilton Anxiety Scale (HAM-A); Modified Overt Aggression Scale (MOAS).

The Columbia Suicide Severity Rating Scale (C-SSRS) was used to assess lifetime and present suicide risk and related features (Suicidal Ideation Severity, Suicidal Ideation Intensity, and Suicidal Behavior).

TLFB and urine sample analysis were used to identify the primary substance of abuse for each patient. Subjects were divided into three macro groups: (1) psychostimulants (e.g., cocaine, amphetamines, synthetic cathinones); (2) depressors (e.g., opioids, alcohol, benzodiazepines); and (3) psychodysleptics (e.g., cannabinoids, psychedelics, dissociatives) [1,25,27,28]. According to their pharmacological profiles, patients were also allocated to a specific substance group: (1) Opioids, Stimulants; (2) Empathogens-Entactogens; (3) Psychedelics; (4) Dissociatives, (5) Cannabinoids; and (6) Depressors. Urine sampling procedure and analysis were described in a previous study [27].

Data collection was carried out anonymously and confidentially; all participants received a detailed explanation of the study design, and written informed consent was obtained. The study was conducted according to the Declaration of Helsinki. Ethics approval was granted by the University of Hertfordshire Health and Human Sciences ECDA, protocol no. aPHAEC1042(03); by the CEI Illes Balears, protocol no. IB 2561/15 P.I.; and by the University “G. d’Annunzio” of Chieti-Pescara, no. 7/09-04-2015. The majorcan local ethics committee also approved the study.

### Data Analysis

Statistical analysis was performed by using IBM SPSS^®^ Statistics software, version 20, and GraphPad 5.0 software for Windows (La Jolla, CA, USA). One-way Analysis of Variance (ANOVA) followed by Tukey’s post-hoc test was used to assess whether there was a significant difference in the intensity of suicidal ideation in life and in the previous month (total score, frequency, duration, and controllability) among subjects who abused the different categories (psychodepressors, psychostimulants, psychodysleptics) or groups (opioids, stimulants, empathogens/enthactogens, psychedelics, cannabinoids, dissociatives) of substances. Regression analysis with Spearman’s correlation values (*ρ*) was applied to assess the significant positive or negative correlations between the intensity of suicidal ideation in life and the previous month (total score, frequency, duration, and controllability) and scores of psychiatric scales. For all tests, a two-tailed *p*-value < 0.05 was considered statistically significant.

## 3. Results

A total of 110 subjects hospitalized in the psychiatric ward of the Can Misses Hospital were enrolled in the study. All the study subjects were diagnosed with substance intoxication at admission. Although the majority of patients indicated to be multiple substance users (*n* = 77, 70.0%), the TFLB test and the urinalysis identified substances of choice for each patient, which were depressors in 17 (15%), stimulants in 44 (40%), and psychodysleptics in 49 (45%) users. When asked about lifetime use of specific groups of substances, stimulant use was disclosed by 74 (32%) patients, followed by cannabis by 68 patients (29%) and depressors by 32 patients (14%). Full results are available in previously published studies [1,27]

The presence of suicide thoughts was evidenced in 35% (*n* = 38) of the sample as to the suicide item of the Hamilton Depression Scale (HAM-D), with 18% (*n* = 20) reporting a severe suicide risk. The assessment of suicidal risk at admission as to the Columbia Suicide Severity Rating Scale (C-SSRS) was performed in 63 subjects of the total sample: 25 (39%) patients were positive for suicide attempts (*n* = 6), suicidal ideation (*n* = 9), or death ideation (*n* = 10). The percentages for each item are reported in Figure 1.

According to the macro-categories of substances, in patients assessed for suicide risk: 5 were psychodysleptics users (1 suicide attempt, 1 suicidal ideation, 3 death ideation), 2 psychodepressors users (both positive for suicidal ideation), and 18 psychostimulants users (5 suicide attempt, 6 suicidal ideations, 7 death ideation).

Suicide risk assessment according to the different groups of substances is shown in Figure 2.

Lifetime Suicide Ideation Intensity (SSI) scores according to C-SSRS were higher in users of psychodepressors compared to psychostimulants, and psychodysleptics in terms of controllability (e.g., the difficulty to control suicidal ideation) (Figure 3).

The same analysis was performed, taking into account the SSI scores in the month before recruitment (Figure 4).

Controllability in the previous month and correlations to substance categories were evaluated in a subset of the study cohort. SSI scores were higher in psychodepressors users compared to the users of other macro-groups of substances. Three subjects used psychodepressors, 16 psychostimulants, and 17 psychodysleptics. Statistical analysis employed One Way ANOVA, followed by Tukey’s post-hoc test, F = 4.488, df = 2, * *p* < 0.05).

Taking into account the substance categories, One Way ANOVA analysis indicated statistically significant differences in lifetime SSI Total (Figure 5A) and Controllability scores (Figure 5D) when evaluating users of opioids and empathogens/entactogens, psychedelics, cannabinoids, dissociatives, and depressors. Lifetime SSI Frequency was higher in opioids users than all the other groups (Figure 5B). Lifetime SSI Duration was higher in users of opioids compared to psychedelics, cannabinoids, and dissociatives users (Figure 5C).

SSI in the previous month was also significantly higher in opioids users compared to stimulants, empathogens/entactogens, psychedelics or cannabinoids users (Figure 6).

Linear regression analysis with Spearman’s correlation value calculation revealed weak negative associations between: all the items of the Lifetime Suicide Ideation Intensity and PANSS positive scores (Total score: R = 0.509, R^2^ = 0.259, *p* = 0.002, *ρ* = −0.485, Frequency: R = 0.435, R^2^ = 0.189, *p* = 0.008, *ρ* = −0.454; Duration: R = 0.533, R^2^ = 0.285, *p* = 0.001; *ρ* = −0.507 and Controllability: R = 0.386, R^2^ = 0.149, *p* = 0.020, *ρ* = −0.398); SSI Frequency and Duration in the previous month and PANNS scores (Frequency: R = 0.379, R^2^ = 0.143, *p* = 0.025, *ρ* = −0.311; Duration: R = 0.356, R^2^ = 0.127, *p* = 0.036, *ρ* = −0.290); Lifetime Suicide Ideation Intensity Total scores, Frequency, Duration and MRS (Total score: R = 0.349, R^2^ = 0.122, *p* = 0.037, *ρ* = −0.246; Frequency: R = 0.363, R^2^ = 0.132, *p* = 0.030, *ρ* = −0.229; Duration R = 0.395, R^2^ = 0.156, *p* = 0.017, *ρ* = −0.263); Last month Suicide Ideation Intensity Frequency and MRS scores (R = 0.394, R^2^ = 0.155, *p* = 0.025, *ρ* = −0.311).

## 4. Discussion

Holiday periods represent a time of risk, excess, and experimentation, especially among young people [29]. In this study, we evaluated the relevance of suicidality—expressed as suicidal ideation, suicidal attempts, and suicidal thoughts—in a cohort of “clubbers” and disco goers in Ibiza, a prosperous “substance market” always up to date for newly developed recreational drugs.

As for suicidal evaluations upon admission, it is interesting to note that 35% of the sample reported suicide risk, with 18% reporting a severe suicide risk as to the suicide item of the HAM-D. The Columbia Suicide Severity Rating Scale showed that 39% of the subjects exhibited suicidality, considered as the positivity of at least one of the following items: suicidal ideation, suicidal attempts, death ideation. When considering the percentage of each of the items suicidal ideation, suicidal attempts, death ideation, and self-harm injuries, 14% showed suicidal ideation (*n* YES = 9; *n* NO = 54); 10% suicidal attempts (*n* YES = 6; *n* NO = 57), 16% suicidal thoughts/fear of death (*n* YES = 10; *n* NO = 53); and 22% self-harm injuries (*n* YES = 14; *n* NO = 49). Moreover, considering the distribution of the items suicidal ideation, suicidal attempts, death ideation, and self-harm injuries in the total number of subjects who showed positivity for them (*n* = 39), suicidal ideation was reported by 23% (*n* = 9), suicidal attempts by 15% (*n* = 6), death ideation by 26% (*n* = 10), and self-injury by 36% (*n* = 14).

It is interesting to note that suicidality levels were also high in the subpopulation that includes adolescents and young adults.

The incidence of suicidality is relevant considering that we assessed a cohort of subjects who do not belong to a clinical context. They accidentally came in contact with clinical psychiatry after substance poisoning in a recreational and vacation context. This group of people has high levels of education and good employment rates and differ from the typical profile of substance abusers [30]. For this reason, this data lead to hypothesize a favoring role of substances in relation to the presence of suicidality. An explanation for this phenomenon relates to changing landscape of substance-using clients. In particular, clubbers and recreational drug users are younger and younger and differ significantly from the “drug addicts” of the past [2,31,32]. Multiple substance abuse was commonly found in the cohort. The combination of psychoactive drugs has health implications [25], generates higher levels of intoxication, ultimately increasing suicidality and fatality [33,34]. Recent studies revealed that suicidality is highly prevalent in individuals with comorbid Major Depressive Disorder and Borderline Personality Disorder (BPD) [35]. Our sample is in line with this morbidity.

In our sample SSI when evaluated in terms of frequency, controllability, and duration in the month before the assessment, was higher in psychodepressors and opioids users in particular. This finding is consistent with other studies [36].

An analysis of 20,917 deaths from opioid poisoning in the United States suggests increased involvement of opioids in intentional overdose and suicide [17,19]. Opioid users have always represented a high suicidal risk category [18] and this phenomenon is also reflected in recent data concerning the new synthetic opioids during the opioid crisis in the United States [37]. The use of psychodepressors could also respond to a hypothesis more linked to the concept of self-medication, in subjects with an underlying anxious substrate. Substances belonging to the class of psychodepressors and opiates are able, at least in the short term, to reduce the amount of anxiety and perceived stress. It should also be considered that some intrinsic characteristics of the substances belonging to these group of substances could favor the development of suicidal elements. Differently, substances belonging to other classes could hypothetically play a protective role, as recently observed for some psychodysleptics.

One novel intervention for resistant depression is ketamine, a dissociative that acts as an antagonist of N-Methyl-D-Aspartate (NMDA), and also able to increase plasmatic BDNF levels [38]. Literature reports also psychedelics as one potentially promising novel intervention for suicidality [39,40,41]. Psychedelics are a class of pharmacological agents, including psilocybin and ayahuasca (a brew which contains N,N-dimethyltryptamine and beta-carboline alkaloids), that induce changes in affect, cognition, and perception, as well as non-ordinary states of consciousness at high doses [42,43]. Interestingly, cross-sectional and longitudinal evidence indicates that lifetime use of psychedelics is associated with lower levels of suicidality [44]. The neurobiological explanation of the phenomenon may rely on evidence indicating that psychedelics induce functional and structural brain changes that have an impact on the modulation of cognition and perception and ultimately reduce suicidality (REF).

In our study, suicide ideation intensity was inversely, although weakly, associated with PANSS positive scores (the higher the PANSS positive, the lower the suicide intensity), thereby showing that the presence of substance-induced psychotic symptoms did not influence the suicide risk. This is in line with the hypothesis of direct influence played by substances on suicidality rather than the emergence of dissociative elements.

Our study presents limitations like (1) the possibility to identify new substances in urine samples remains complex and limited. Although we performed rigorous screenings, the match between self-reported drug use and objective data is still far from reliable; (2) the systematization that we employed based on groups and categories is probably not ideal, as use of multiple substances was the predominant behavior among our study participants; (3) the long-term drug effects in terms of suicide risk are difficult to assess without follow-up examinations; (4) the sample size of subjects that we evaluated for suicide risk was somehow suboptimal, as well as the distribution in subgroups, limiting the statistical power of the study.

## 5. Conclusions

In this study, conducted on a sample of young adults with a high level of education and a good socioeconomic status, high scores of suicidality correlated with the presence, in the month before the assessment, of suicide attempts, suicidal ideation, thoughts of death, and self-harming behavior. These symptoms prevailed in users of psychodepressors and specifically opioids. This finding calls for urgent psychoeducational and preventive strategies targeted to this subgroup of users. The presence of high levels of suicidality did not specifically correlate with undergoing significant psychopathological conditions, thereby indicating an independent association between the use of psychoactive substances and suicidality. Instead, our data favor the idea of a possible role played by impulsivity and the loss of control in the propensity to and a higher risk of suicide.

## Figures and Tables

**Figure 1 brainsci-11-00490-f001:**
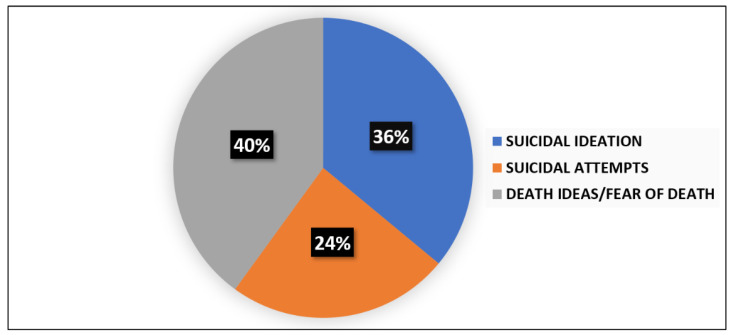
Prevalence of suicidal ideation, suicide attempts, and death ideation among patients with high risk of suicidality at entry evaluation.

**Figure 2 brainsci-11-00490-f002:**
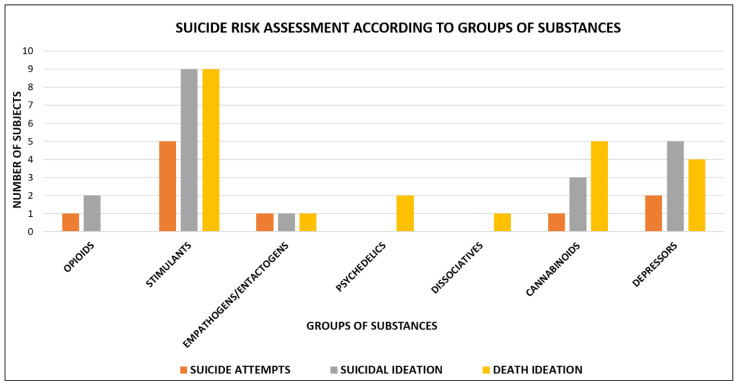
Suicide risk assessment according to groups of substances: opioids users = 3 (1 suicide attempt, 2 suicidal ideation); stimulants users = 23 (5 suicide attempts, 9 suicidal ideation and 9 death ideation); empathogens/entactogens users = 3 (1 suicide attempt, 1 suicidal ideation and 1 death ideation); psychedelics users = 2 (2 death ideation); dissociatives users = 1 (1 death ideation); cannabinoids users = 9 (1 suicide attempt, 3 suicidal ideation and 5 death ideation); depressors users = 11 (2 suicide attempt, 5 suicidal ideation and 4 death ideation).

**Figure 3 brainsci-11-00490-f003:**
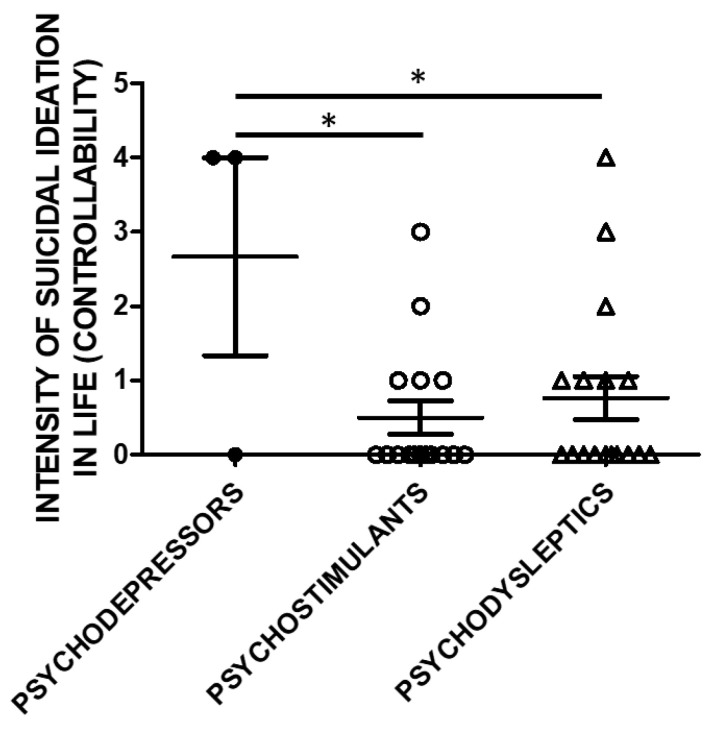
Suicidal Ideation Intensity (SSI; Controllability) and categories of substances of use. The history of controllability and its correlation with substance categories was evaluated in a subset of the study cohort. Three used psychodepressors, 16 psychostimulants, and 17 psychodysleptics. Statistical analysis employed One Way ANOVA, followed by Tukey’s post-hoc test, F = 4.299, df = 2, * *p* < 0.05).

**Figure 4 brainsci-11-00490-f004:**
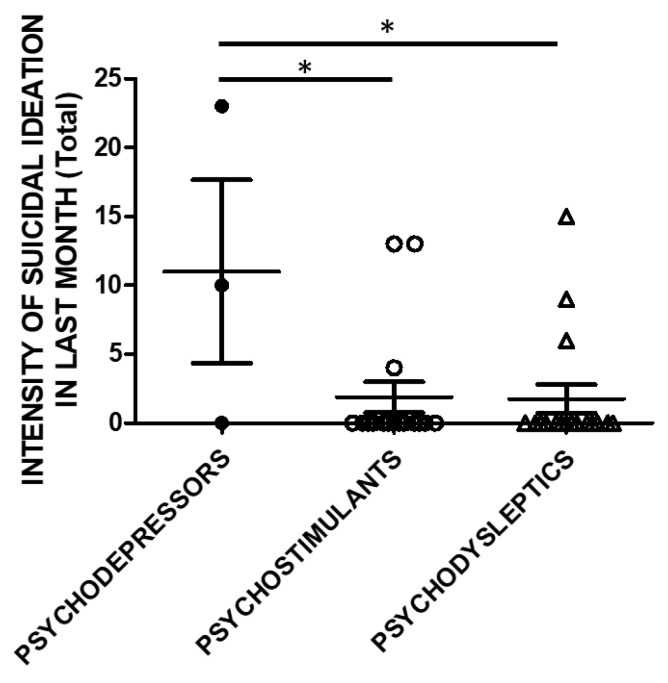
SSI in the previous month (Total score) and categories of substances. Three subjects used psychodepressors, 16 psychostimulants, and 17 psychodysleptics. Statistical analysis employed One Way ANOVA, followed by Tukey’s post-hoc test, F = 4.488, df = 2, * *p* < 0.05).

**Figure 5 brainsci-11-00490-f005:**
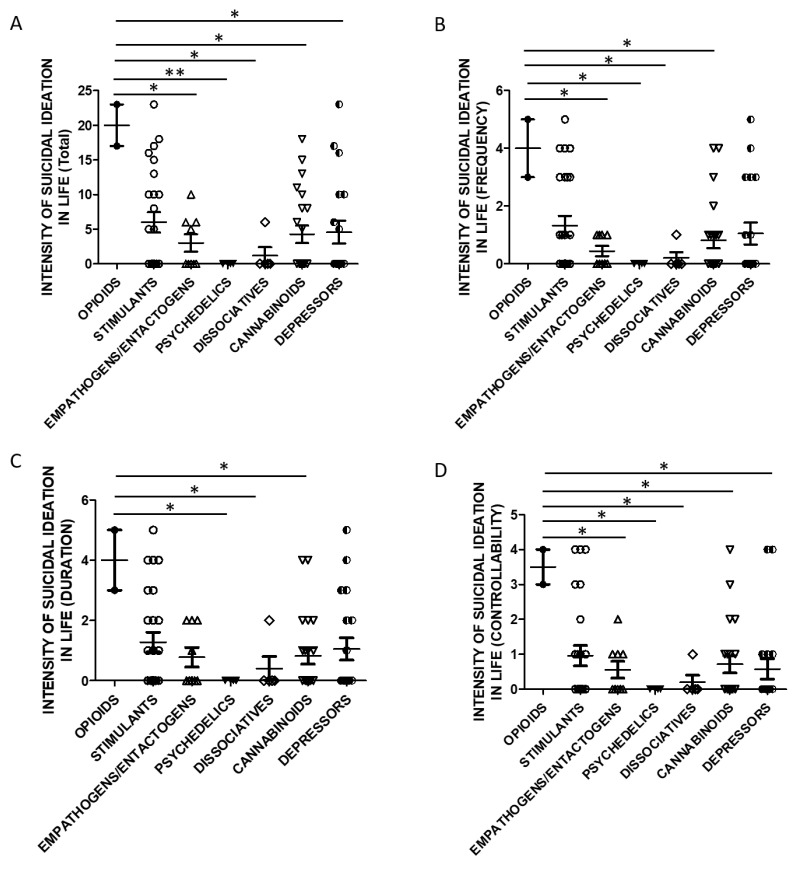
SSI in life and groups of substances (**A**)-Total Score, opioids *n* = 2, stimulants *n* = 25, empathogens/entactogens *n* = 9; psychedelics *n* = 4, dissociatives *n* = 5, cannabinoids *n* = 22, depressors *n* = 19. One Way ANOVA, followed by Tukey’s post-hoc test, F = 2.902, df = 6, * *p* < 0.05, ** *p* < 0.01; (**B**)-Frequency, opioids *n* = 2, stimulants *n* = 25, empathogens/enthactogens *n* = 9; psychedelics *n* = 4, dissociatives *n* = 5, cannabinoids *n* = 22, depressors *n* = 19. One Way ANOVA, followed by Tukey’s post-hoc test, F = 2.595, df = 6, * *p* < 0.05; (**C**)-Duration, opioids *n* = 2, stimulants *n* = 25, empathogens/entactogens *n* = 9; psychedelics *n* = 4, dissociatives *n* = 5, cannabinoids *n* = 22, depressors *n* = 19, One Way ANOVA, followed by Tukey’s post-hoc test, F = 2.248, df = 6, * *p* < 0.05; (**D**)-Controllability opioids *n* = 2, stimulants *n* = 25, empathogens/enthactogens *n* = 9; psychedelics *n* = 4, dissociatives *n* = 5, cannabinoids *n* = 21, depressors *n* = 19, One Way ANOVA, followed by Tukey’s post-hoc test, F = 2.440, df = 6, * *p* < 0.05).

**Figure 6 brainsci-11-00490-f006:**
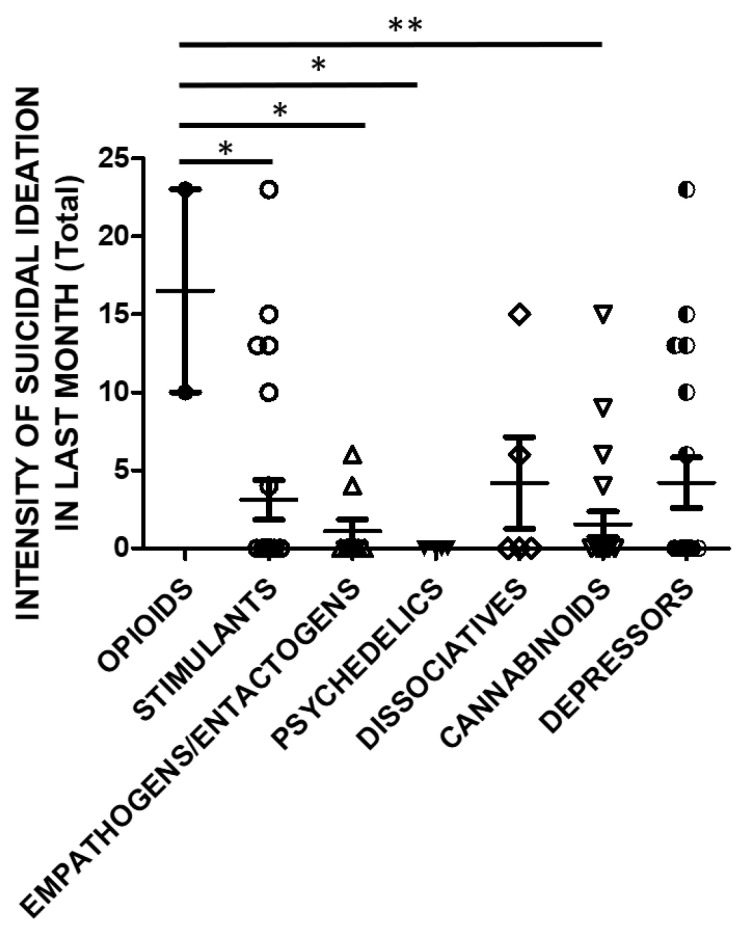
SSI in last month (Total Score) and groups of substances (opioids *n* = 2, stimulants *n* = 25, empathogens/entactogens *n* = 9; psychedelics *n* = 4, dissociatives *n* = 5, cannabinoids *n* = 22, depressors *n* = 19; One Way ANOVA, followed by Tukey’s post-hoc test, F = 2.754, df = 6, * *p* < 0.05, ** *p* < 0.01).

## Data Availability

The data presented in this study are available on request from the corresponding author.

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
