# Peer review of "Suicidal Behavior and Club Drugs in Young Adults"

_brainsci, 2021, doi:10.3390/brainsci11040490_

Round 1

Reviewer 1 Report

The manuscript entitled "Suicidal behavior and club drugs in young
adults" reports research results  which indicate that there are
significant relationships between suicidal behaviors and the use of
various psychoactive substances. The subject matter of the article and
the results it presents concern very important problems in the area of
public health. The article deserves to be published, but it has to be
properly revised. In particular, I find the use of the three scales:
PANSS, MRS  and HAM-D methodologically incorrect, because these
instruments are intended for use in patients with schizophrenia (PANSS),
manic states (MRS) and depression (HAM-D), and not in patients with
symptoms of substance abuse.

Author Response

The manuscript entitled "Suicidal behavior and club drugs in young adults" reports research results which indicate that there are significant relationships between suicidal behaviors and the use of various psychoactive substances. The subject matter of the article and the results it presents concern very important problems in the area of public health. The article deserves to be published, but it has to be properly revised. In particular, I find the use of the three scales: PANSS, MRS and HAM-D methodologically incorrect, because these instruments are intended for use in patients with schizophrenia (PANSS), manic states (MRS) and depression (HAM-D), and not in patients with symptoms of substance abuse.

We thank the reviewer for his suggestions. We have now explained the reason of this choice in the method section and report this issue as a possible limitation in the discussion of the revised version of the manuscript. However, we would like to report that, according to the most recent studies, schizophrenia, manic states, and depression resulted to be highly comorbid with substance use, in percentages varying from 22% to 89% (Hunt GE, Malhi GS, Lai HMX, Cleary M. Prevalence of comorbid substance use in major depressive disorder in community and clinical settings, 1990-2019: Systematic review and meta-analysis. J Affect Disord. 2020 Apr 1;266:288-304; Martinotti G, De Risio L, Vannini C, Schifano F, Pettorruso M, Di Giannantonio M. Substance-related exogenous psychosis: a postmodern syndrome. CNS Spectr. 2021 Feb;26(1):84-91; Hunt GE, Large MM, Cleary M, Lai HMX, Saunders JB. Prevalence of comorbid substance use in schizophrenia spectrum disorders in community and clinical settings, 1990-2017: Systematic review and meta-analysis. Drug Alcohol Depend. 2018 Oct 1;191:234-258).

Reviewer 2 Report

Title: “Suicidal behavior and club drugs in young adults“

In this study, the authors investigated suicidality prevalence among drug users and evaluated differences in suicide ideation, taking into account substance categories and the association of suicide ideation intensity with other psychiatric symptoms. The authors studied subjects admitted to the Can Misses Hospital's psychiatry ward in Ibiza. These subjects were recruited during summer openings of local nightclubs for four consecutive years starting in 2015. In particular, the main inclusion criterium was an intake of psychoactive substances during the previous 24 hours. The authors used the Columbia Suicide Severity Rating Scale (C-SSRS) to assess suicide risk. The authors found that suicidality was present in 39% of the study cohort. In addition, Suicide Ideation Intensity overall and in the previous month was higher in users of opioids and in general of psychodepressors compared to psychostimulants or psychodysleptics. Furthermore, the authors found that suicidality was not correlated with alterations in any of the major psychopathological scales employed to assess the psychiatric background of the study subjects. The presence of high levels of suicidality did not specifically correlate with any major symptom indicative of previous or ongoing psychopathological alterations. The authors claim that these findings support the idea that impulsivity and loss of self-control are the major determinants of the increased suicidality irrespectively of any major ongoing psychiatric background.

General comment:  This study is interesting, quite well organized and written. The aim of this work is also clearly expressed as “to assess psychopathological and suicidal features associated with substance use”, while “suicidality” was expressed through suicidal ideation, suicide attempts, and death ideation according to The Columbia Suicide Severity Rating Scale (C-SSRS).

More specifically, the authors claim that: “The study also aimed explicitly to (1) identify the suicidality prevalence among drugs misusers, (2) evaluate differences in Suicide Ideation Intensity (SSI; occurring through life and in the month before recruitment) among users of different substances, (3) evaluate associations of SSI (lifetime and in the last month) with psychopathological features assessed by the Positive and Negative Symptoms Scale (PANSS), Mania Rating Scale (MRS), and Hamilton Depression Scale (HAM-D) scores.”

Nevertheless, the value of the statistical analysis is not always clear and should be clarified to support the main results. Similarly, plots are not always clear and should be improved. In addition, the “predictive power” of the provided work should be better explained. Finally, the main conclusions should be better presented also with reference to the current state of the art.

Some specific comment:

Section “Material and Methods

Lines: “The Columbia Suicide Severity Rating Scale (C-SSRS) was used to assess lifetime and present suicide risk and related features (Suicidal Ideation Severity, Suicidal Ideation Intensity, and Suicidal Behavior).”

*) The authors should discuss in depth the use of this scale together with its main limitations within the “Discussion” section of this work.

Lines: “Data collection was carried out anonymously and confidentially; all participants received a detailed explanation of the study design, and written informed consent was obtained. The study was conducted according to the Declaration of Helsinki. Ethics approval was granted by the University of Hertfordshire Health and Human Sciences ECDA, protocol no. aPHAEC1042(03); by the CEI Illes Balears, protocol no. IB 2561/15 P.I.; and by the University "G. d'Annunzio" of Chieti‐Pescara, no. 7/09‐04‐2015. The majorcan local ethics committee also approved the study.”

*) Hopefully, from an ethical point of view, a correct information on the main consequences of the substances use has been delivered to all participants. Could the authors comment this point ?

Section: “Results”

Figure 1. Prevalence of use of specific groups of substances in subjects aged ≤ 25 years.

*) The quality of this figure should be enhanced, for instance improving the background colour.

Lines: “When asked about lifetime use of specific groups of substances, stimulant use was

disclosed by 74 (32%) patients, followed by cannabis by 68 patients (29%) and depressors

by 32 patients (14%). Full results are available in previously published studies [1, 27]”

*) The authors should better explain within the “Discussion” section the novelty and the value of this new work with reference to the previous works cited in [1, 27]

Figure 2. prevalence of suicidal ideation, suicide attempts, and death ideation among patients with high risk of suicidality at entry evaluation

*) The quality of this plot should be improved. See the previous comment about Figure 1

Figure 3. suicidality and substances of use.

*) This plot is interesting. However, it is unclear. The authors should provide a better version of this figure together with complete and meaningful labels and caption.

Lines: “Linear regression analysis with Spearman’s correlation value calculation showed a

negative association between all the items of the Lifetime Suicide Ideation Intensity and

PANSS positive scores (Total score: R=0.509, p=0.002, = -0.485, Frequency: R=0.435,

p=0.008, = -0.454; Duration: R=0.533, p=0.001; = -0.507 and Controllability: R=0.386,

p=0.020, = -0.398). In addition to this, SSI Frequency and Duration in the previous month

were negatively associated with PANNS scores (Frequency: R=0.379, p=0.025, = -0.311;

Duration: R=0.356, p=0.036, = -0.290). Furthermore, a significant negative correlation between Lifetime Suicide Ideation Intensity Total scores, Frequency, Duration and MRS

scores was found (Total score: R=0.349, p=0.037, = -0.246; Frequency: R=0.363, p=0.030, =

-0.229; Duration R=0.395, p=0.017, = -0.263). Last month Suicide Ideation Intensity Frequency was negatively associated with MRS scores (R=0.394, p=0.025, = -0.311)"

*) The authors should better clarify whether the values of R index are really meaningful, since they seem to be quite low. In particular, a calculation of the R-square standard index could make this point particularly clear. Please clarify.

Sections “Discussion”, “Conclusions”

Lines: “In this study, we evaluated the relevance of suicidality - expressed as suicidal ideation, suicidal attempts, and suicidal thoughts - in a cohort of "clubbers" and disco goers in Ibiza, a prosperous 'substance market' always up to date for newly developed recreational drugs.”

*) In this work suicidal ideation, suicidal attempts, and suicidal thoughts have been related only to the use of different kind of substances. However, perhaps, this kind of suicidal thoughts could be related also (and perhaps mainly) to the current “psychological state” of people. Indeed, perhaps, also the use of substance could be explained in these “psychological” way, since the causes could be the loss of life sense due the loss of the religious sense, the failure of their future prospects in life, major family tragedies together with their severe psychological effects, etc,etc. The authors should comment these points within the “Discussion” section. Similarly, the novelty of this work should be better empathized with respect to the current state of the art.

Author Response

Reviewer 2

In this study, the authors investigated suicidality prevalence among drug users and evaluated differences in suicide ideation, taking into account substance categories and the association of suicide ideation intensity with other psychiatric symptoms. The authors studied subjects admitted to the Can Misses Hospital's psychiatry ward in Ibiza. These subjects were recruited during summer openings of local nightclubs for four consecutive years starting in 2015. In particular, the main inclusion criterium was an intake of psychoactive substances during the previous 24 hours. The authors used the Columbia Suicide Severity Rating Scale (C-SSRS) to assess suicide risk. The authors found that suicidality was present in 39% of the study cohort. In addition, Suicide Ideation Intensity overall and in the previous month was higher in users of opioids and in general of psychodepressors compared to psychostimulants or psychodysleptics. Furthermore, the authors found that suicidality was not correlated with alterations in any of the major psychopathological scales employed to assess the psychiatric background of the study subjects. The presence of high levels of suicidality did not specifically correlate with any major symptom indicative of previous or ongoing psychopathological alterations. The authors claim that these findings support the idea that impulsivity and loss of self-control are the major determinants of the increased suicidality irrespectively of any major ongoing psychiatric background. General comment: This study is interesting, quite well organized and written. The aim of this work is also clearly expressed as “to assess psychopathological and suicidal features associated with substance use”, while “suicidality” was expressed through suicidal ideation, suicide attempts, and death ideation according to The Columbia Suicide Severity Rating Scale (C-SSRS). More specifically, the authors claim that: “The study also aimed explicitly to (1) identify the suicidality prevalence among drugs misusers, (2) evaluate differences in Suicide Ideation Intensity (SSI; occurring through life and in the month before recruitment) among users of different substances, (3) evaluate associations of SSI (lifetime and in the last month) with psychopathological features assessed by the Positive and Negative Symptoms Scale (PANSS), Mania Rating Scale (MRS), and Hamilton Depression Scale (HAM-D) scores.” Nevertheless, the value of the statistical analysis is not always clear and should be clarified to support the main results. Similarly, plots are not always clear and should be improved. In addition, the “predictive power” of the provided work should be better explained. Finally, the main conclusions should be better presented also with reference to the current state of the art.

We thank the Reviewer for his/her very careful evaluation of our manuscript and for providing us valuable comments and suggestions to improve it. With respect to the issues raised in his/her general comment, we fully deal with them by providing a “point by point” reply to his/her specific comments below.

Some specific comment:

Section “Material and Methods

Lines: “The Columbia Suicide Severity Rating Scale (C-SSRS) was used to assess lifetime and present suicide risk and related features (Suicidal Ideation Severity, Suicidal Ideation Intensity, and Suicidal Behavior).”

*) The authors should discuss in depth the use of this scale together with its main limitations within the “Discussion” section of this work.

We thank the reviewer for his/her suggestion. As requested, a part on the use and limitations of this scale has been added in the Discussion section of the revised version of the manuscript.

Lines: “Data collection was carried out anonymously and confidentially; all participants received a detailed explanation of the study design, and written informed consent was obtained. The study was conducted according to the Declaration of Helsinki. Ethics approval was granted by the University of Hertfordshire Health and Human Sciences ECDA, protocol no. aPHAEC1042(03); by the CEI Illes Balears, protocol no. IB 2561/15 P.I.; and by the University "G. d'Annunzio" of Chieti‐Pescara, no. 7/09‐04‐2015. The majorcan local ethics committee also approved the study.”

*) Hopefully, from an ethical point of view, a correct information on the main consequences of the substances use has been delivered to all participants. Could the authors comment this point?

All the participants received an educational session inside the Can Misses Hospital. This was a clinical protocol specifically dedicated to substance users.

Section: “Results”

Figure 1. Prevalence of use of specific groups of substances in subjects aged ≤ 25 years.

*) The quality of this figure should be enhanced, for instance improving the background colour.

We thank the Reviewer for his/her comment. As requested, Figure 1 has been revised improving its quality and, in particular, the background colour.

Lines: “When asked about lifetime use of specific groups of substances, stimulant use was

disclosed by 74 (32%) patients, followed by cannabis by 68 patients (29%) and depressors

by 32 patients (14%). Full results are available in previously published studies [1, 27]”

*) The authors should better explain within the “Discussion” section the novelty and the value of this new work with reference to the previous works cited in [1, 27]

As suggested, we have now explained the novel points of this study with respect to previous contributions.

Figure 2. prevalence of suicidal ideation, suicide attempts, and death ideation among patients with high risk of suicidality at entry evaluation

 *) The quality of this plot should be improved. See the previous comment about Figure 1

We thank the Reviewer for his/her comment. As requested, Figure 2 has been revised improving its quality.

Figure 3. suicidality and substances of use.

*) This plot is interesting. However, it is unclear. The authors should provide a better version of this figure together with complete and meaningful labels and caption.

We thank the Reviewer for his/her comment. An improved version of Figure 3 with complete labels and caption has been provided in the revised version of the manuscript.

Lines: “Linear regression analysis with Spearman’s correlation value calculation showed a

negative association between all the items of the Lifetime Suicide Ideation Intensity and

PANSS positive scores (Total score: R=0.509, p=0.002, r= -0.485, Frequency: R=0.435,

p=0.008, r= -0.454; Duration: R=0.533, p=0.001; r= -0.507 and Controllability: R=0.386,

p=0.020, r= -0.398). In addition to this, SSI Frequency and Duration in the previous month

were negatively associated with PANNS scores (Frequency: R=0.379, p=0.025, r= -0.311;

Duration: R=0.356, p=0.036, r= -0.290). Furthermore, a significant negative correlation between Lifetime Suicide Ideation Intensity Total scores, Frequency, Duration and MRS

scores was found (Total score: R=0.349, p=0.037, r= -0.246; Frequency: R=0.363, p=0.030, r=-0.229; Duration R=0.395, p=0.017, r= -0.263). Last month Suicide Ideation Intensity Frequency was negatively associated with MRS scores (R=0.394, p=0.025, r= -0.311)"

*) The authors should better clarify whether the values of R index are really meaningful, since they seem to be quite low. In particular, a calculation of the R-square standard index could make this point particularly clear. Please clarify.

We thank the Reviewer for this valuable and helpful comment which allowed us to better interpretate our Results. Indeed, as requested, we calculated R-square standard indexes (values have been also included in the Results section of the revised version of the manuscript), which definitely pointed towards weak associations. This has been specified in the Results and Discussion sections of the revised version of the manuscript.

Sections “Discussion”, “Conclusions”

Lines: “In this study, we evaluated the relevance of suicidality - expressed as suicidal ideation, suicidal attempts, and suicidal thoughts - in a cohort of "clubbers" and disco goers in Ibiza, a prosperous 'substance market' always up to date for newly developed recreational drugs.”

*) In this work suicidal ideation, suicidal attempts, and suicidal thoughts have been related only to the use of different kind of substances. However, perhaps, this kind of suicidal thoughts could be related also (and perhaps mainly) to the current “psychological state” of people. Indeed, perhaps, also the use of substance could be explained in these “psychological” way, since the causes could be the loss of life sense due the loss of the religious sense, the failure of their future prospects in life, major family tragedies together with their severe psychological effects, etc,etc. The authors should comment these points within the “Discussion” section. Similarly, the novelty of this work should be better empathized with respect to the current state of the art.

In the discussion section we have now reported and discussed the role of a possible “psychological state” that could have represented a common ground in relation to both suicide ideation and substance use
